# Sedimentary DNA is a promising indicator of the abundance of marine benthos: Insights from the burrowing decapod *Upogebia major*

**Kyosuke Kitabatake**[1,2,3*], **Kentaro Izumi**[2], **Natsuko Ito-Kondo**[4], **Kenji Okoshi**[1,3]

1 Graduate School of Science, Toho University, Funabashi, Chiba, Japan, 2 Faculty and Graduate School of Education, Chiba University, Chiba, Japan, 3 Department of Science, Tokyo Bay Ecosystem Research Center, Toho University, Funabashi, Chiba, Japan, 4 Biodiversity Division, National Institute for Environmental Studies, Tsukuba, Ibaraki, Japan

* kyosuke.kitabatake@gmail.com

## Abstract

*Upogebia major* (De Haan 1841) is a decapod widely distributed in tidal flats in East Asia and Russia and is a dominant species in some regions. Nevertheless, because the inhabiting deep burrows exceeding a depth of 2 m, conducting quantitative surveys is extremely challenging. These challenges are not unique to this species but are also common to infaunal marine benthos. Therefore, this study focuses on environmental DNA (eDNA). eDNA consists of DNA fragments present in water and sediments and serves as an indicator of the presence and abundance of organisms. In particular, sedimentary DNA (sedDNA) is highly concentrated compared with that found in water. When sedDNA is used as an indicator of abundance, it is essential to first comprehensively understand the relationship between abundance and sedDNA concentration. Hence, the number of burrow openings in *U. major* was considered as abundance, and the relationship between the seasonal variation in abundance and sedDNA concentration was investigated in three tidal flats with different burrow distributions and topographies. During the growth period of this species in stable bottom environments such as bag-shaped lagoons, $10^5$ copies/g sediment of sedDNA was detected in burrow-rich areas, which was significantly higher than in other areas. This indicated a correlation between abundance and concentration. However, it was found that events such as spawning, strong wave action, and changes in activity due to increase in water temperature induced fluctuations in concentration. Therefore, while the sedDNA concentration has the potential to reflect abundance, it is susceptible to biological and physical influences; hence, caution is required regarding the location and timing of surveys. This study sheds light on a fraction of the relationship between *U. major* abundance and sedDNA concentration, offering initial insights into the applicability of sedDNA analysis for estimating the abundance of various marine benthos.

## Introduction

Insights into the distribution and abundance of organisms are paramount in various fields, including ecology, biodiversity conservation, and fisheries [1–3]. Notably, marine benthos

**Data availability statement:** All relevant data are within the manuscript and its Supporting Information files.

**Funding:** All relevant data are within the paper and its Supporting Information files.

**Competing interests:** The authors have declared that no competing interests exist.

inhabiting the intertidal zone contribute not only to material cycling and environmental purification [4–6] but also represent valuable assets in the field of fisheries [7, 8]. Consequently, the imperative need for quantitative evaluations remains unassailable.

The *Upogebia* genus is a cluster of decapod crustaceans distributed globally, encompassing approximately 280 species. *Upogebia major* (De Haan 1841) is a large mud shrimp that can reach a total length of approximately 10 cm. It is widely distributed in the sandy and muddy tidal flats of the Japanese Archipelago, the Korean Peninsula, the Shandong Peninsula, and the Russian coast facing the Sea of Japan [9–13]. Although predominantly found in East Asia, it has also been documented in San Francisco Bay. The population in San Francisco Bay is likely an alien species, with [14] suggesting that larvae introduced via ballast water a highly likely to have been established there. According to [15], it has been documented that Chinese and Korean individuals of *U. major* are distributed in Japan. The possibility that continental individuals were introduced to Japan along with the anthropogenic movement of the Asari clam, *Ruditapes philippinarum* is being investigated. In certain regions of Japan and South Korea, *U. major* is considered the dominant species in tidal flats [10,11,16], with estimated population densities exceeding 200 individuals/m$^2$ in some areas [16, 17]. Furthermore, upogebiid shrimps function as filter feeders, predominantly consuming phytoplankton in seawater, thus contributing to coastal water purification. According to [18], in the Lagoon of Grado (Northern Adriatic), all inflowing seawater is filtered by upogebiid shrimps. The most prominent characteristic of *U. major* is the creation of large Y-shaped burrows exceeding a depth of 2 m [9,13]. These burrows are sometimes evenly distributed across the entire tidal flat; however, they are often concentrated in specific areas within the tidal flat (Kitabatake, personal observation), which are referred to as "burrow-rich areas" in this study. The interior of the burrow is fortified with materials, such as mud and mucus, rendering it resistant to collapse. Owing to its robust and large burrows, as well as the diverse array of symbiotic organisms inhabiting its body, *U. major* is widely recognized as a notable host [19–21]. Additionally, it should be noted that larger individuals of *U. major* are commonly collected as seafood in the Okayama and Kumamoto prefectures of Western Japan and South Korea [16,22–24], whereas smaller specimens are primarily utilized as fishing bait in Japan (Kitabatake, personal observation).

This species, while exhibiting a wide distribution across East Asia as a generalist, crafts some of the most substantial burrow structures within tidal flats, giving rise to extensive subterranean voids. The presence and behavior of *U. major* significantly impact the physical, chemical, and biological aspects of tidal flat ecosystems and coastal environments. It is not an exaggeration to state that its scale is the largest among the benthos in tidal flats. Moreover, in certain regions, this species is deeply intertwined with people's lives as a source of seafood and a component of leisure activities. Hence, precise evaluation of species distribution and abundance, coupled with dedicated monitoring endeavors, is imperative. Quantitative assessments of upogebiid shrimps have predominantly involved counting the number of burrows and collecting live individuals [17,25,26]. Nonetheless, the former lacks clarity regarding the presence of live organisms within the burrows. Furthermore, an accurate assessment becomes challenging when the burrows are filled with sediment or obscured. However, the latter, which involves the collection of this species dwelling in deep burrows, is not easily executed; thus, variations in collection capabilities can impact the survey results. Both methods require significant manual labor and time, and due to issues with accuracy, quantitative surveys have rarely been conducted in recent years. Hence, there is an urgent need to develop a facile and precision-centric approach. These challenges are not limited to this species but are common to infaunal marine benthos.

Amid these challenges, recent years have witnessed the feasibility of conducting convenient and expeditious field surveys by harnessing environmental DNA (eDNA), which

comprises the DNA fragments found in water and sediments. eDNA analysis is cost-effective and requires small sample volumes of approximately 1 L for water and 1 g or less for sediment samples. In particular, eDNA analysis has demonstrated high efficacy for estimating the presence or absence of organisms [27, 28]. When the eDNA of the target organism is detected, it can be determined as "present." Utilizing these data allows for the estimation of the distribution range of an organism [29–31]. However, the applicability of eDNA analysis to quantitative estimations, such as existing abundance, remains variable. Previous studies have observed a robust correlation between the number of Cyprinidae fish, such as *Cyprinus carpio* and eDNA concentration in experimental tanks and mesocosms [32, 33]. In natural environments, a correlation has been identified between the quantitative abundance of lake trout *Salvelinus namaycush* and Walleye *Sander vitreus* and their respective eDNA [34, 35]. However, this relationship is not always pronounced, particularly in natural environments [36]. When transitioning from experimental settings to field conditions, several factors can obscure the relationship between eDNA and abundance [37]. For instance, juveniles of the bluegill *Lepomis macrochirus* often exhibit a higher rate of eDNA release than adults [38], which could potentially result in significant disparities in abundance estimates based on the balance between adults and juveniles. Furthermore, in sockeye salmon *Oncorhynchus nerka*, an increase in eDNA release has been reported during the spawning season, coinciding with the release of gametes and hatching of larvae, leading to alterations in concentration and detection range owing to widespread dispersion [39]. On the other hand, while most previous eDNA studies have focused on water samples, recent studies have revealed that sedimentary DNA (sedDNA) is present in higher concentrations than eDNA in water. SedDNA exhibits excellent preservation owing to its reduced degradation rates as it adsorbs onto clay minerals [40, 41]. Furthermore, in the case of benthic organisms such as *U. major*, there is a higher possibility that eDNA derived from this species exists in higher concentrations in sediments than in water, suggesting strong compatibility with sedDNA analysis in such cases. However, research has traditionally focused on important fishery and invasive species. Few sedDNA studies have been conducted on infaunal marine benthos, such as this species, which are difficult to quantify. Additionally, owing to the historical focus of eDNA research on terrestrial environments, studies targeting marine environments are in the developmental stage.

*U. major* is not only widely distributed but also burrows deeper into the sediment than many other tidal flat organisms. Developing a tool to estimate the abundance of *U. major* using sedDNA could potentially be applied to other infaunal marine benthos. To achieve this, it is first necessary to evaluate the correlation between the abundance of this species and sedDNA concentration under natural environmental conditions. Specifically, it is important to investigate whether areas with higher abundance have higher sedDNA concentrations and whether seasonal variations and topography influence sedDNA concentrations. However, as mentioned earlier, due to the difficulty in collecting *U. major*, directly comparing abundance and sedDNA concentrations is challenging.

In this study, the number of burrow openings of *U. major* was considered as an indicator of abundance, and the following two aspects were primarily investigated across three tidal flats with differing topography: 1. We examined whether there were seasonal differences in sedDNA concentration between burrow-rich areas (i.e., areas with high abundance) and areas without burrows within the same tidal flat. 2. We evaluated the relationship between seasonal variations in the number of burrow openings and sedDNA concentrations in tidal flats without distinct burrow-rich areas (i.e., tidal flats where burrows are distributed throughout the entire flat). In case 1, if sedDNA concentrations were significantly higher in burrow-rich areas, and in case 2, if comparable sedDNA concentrations were detected throughout the entire area, these findings were interpreted in this study as evidence of a correlation between

abundance and sedDNA concentrations. Furthermore, we investigated whether there were any seasons in which no correlation was observed and examined the most suitable seasons and geographical conditions for analyzing sedDNA of this species.

## Materials and methods

### Study sites

Field surveys were conducted on three tidal flats in Japan (Fig 1), where the distribution of *U. major* has already been confirmed by [15]. The study sites were Lake Akkeshi in Hokkaido, the artificial tidal flat for *R. philippinarum* culture located within the Mangoku-ura Lagoon, Miyagi (hereinafter referred to as Mangoku-ura), and Sanbanze, Chiba. These study sites are registered in the Global Biodiversity Information Facility (https://doi.org/10.15468/wmdf6k). The characteristics of each study site are summarized in Table 1.

Lake Akkeshi (Fig 1a) is a bag-shaped lagoon fed by several small rivers, including the Bekanbeushi River. It is a semi-enclosed bay-like water body characterized by relatively calm wave conditions. Tidal flats at the mouth of the lake and along the lakeshore are used for

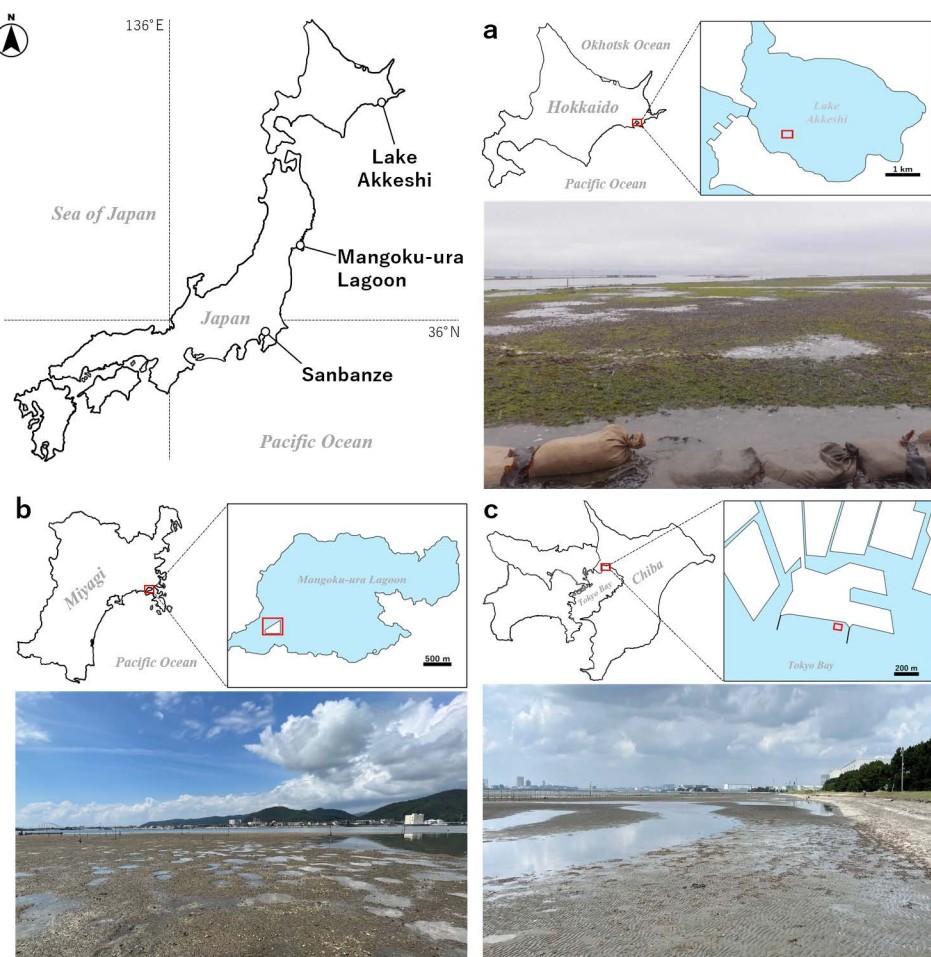

**Fig 1. Map of the study sites for *U. major* distribution assessment and sampling of sediment.** Red squares indicate survey sites. (a) Lake Akkeshi, (b) Mangoku-ura Lagoon, (c) Sanbanze. The images captured in the photographs represent the local conditions at low tide in each case.

**Table 1. The topographical features, the degree of anthropogenic impact, and the distribution patterns of burrows in the study sites. A: Applicable, N/A: Not applicable.**

| Geographical coordinates | | Topographical characteristics | Presence of large inflowing rivers | Anthropogenic impact | Presence of burrow-rich area |
|---|---|---|---|---|---|
| Lake Akkeshi | 44.0425°N; 144.1704°E | Bag-shaped lagoon | A | Rare | N/A |
| Mangoku-ura | 38.4185°N; 141.3820°E | Bag-shaped lagoon | N/A | Rare | A |
| Sanbanze | 35.6709°N; 139.9689°E | Foreshore | N/A | Common | A |

harvesting Asari clams, and the sediment of the tidal flats is an artificial mixture of crushed gravel with the original natural sediment. The survey in this study was conducted at a tidal flat at the mouth of the lake. This site is notable for being accessible only by boat and is designated as a no-fishing zone, contributing to its minimal anthropogenic impact. In the tidal flats located within Lake Akkeshi, *U. major* is one of the dominant species [21,42], and preliminary surveys have confirmed that burrows are distributed throughout the entire area. Mangoku-ura (Fig 1b) was constructed from 2013 to 2016 after the Great East Japan Earthquake of 2011 [43]. The earthquake resulted in approximately 80 cm of ground subsidence around Mangoku-ura [44], causing the former intertidal zone, which served as a fishery for Asari clams, to transition into a subtidal zone [45]. The creation of this tidal flat was initiated to aid fishery recovery. The tidal flat is located within a bag-shaped lagoon and, similar to the survey site at Lake Akkeshi, is accessible only by boat. Consequently, it was characterized by minimal changes in the bottom environment. In addition, owing to the absence of large inflowing rivers into Mangoku-ura Lagoon, the influence of freshwater is minimal. Sanbanze (Fig 1c) is an artificially constructed tidal flat (foreshore tidal flat) created for recreational purposes. The environment experiences waves with an average height of approximately 0.5 to 1 m throughout the year, leading to widespread ripples on the seabed [46]. Furthermore, it is used as a clam-digging area, with a notable influx of tourists during the Asari clam collection season, particularly from March to May. Preliminary surveys revealed the presence of *U. major* burrow-rich areas in both Mangoku-ura and Sanbanze.

## Estimation of seasonal abundance of *U. major* and evaluation of burrow opening morphology

The abundance of *U. major* was estimated based on the number of burrow openings. A survey of the number of burrow openings utilized a 50 × 50 cm quadrat, within which the number of openings was counted. In Lake Akkeshi, where burrows are widely distributed throughout the tidal flat, quadrats were placed randomly. In contrast, at Mangoku-ura and Sanbanze, quadrats were placed within burrow-rich areas. Five quadrat surveys were conducted at each site. It is worth noting that this species exhibits two openings per individual burrow; hence, its abundance was estimated by considering half the number of burrow openings.

In addition, the longest diameter of the burrow openings was measured, and their shape was observed. Measurement of longest diameters was performed through image analysis using ImageJ (ver. 1.54c software, https://imagej.net/ij/). The morphology of the burrows was assessed by examining the shape of the openings and the presence of decorations. Burrowing marine benthos sometimes decorate their burrows using surrounding sediments, shells, and other materials. Therefore, it is necessary not only to confirm the shape of the openings, but also to confirm the presence or absence of decorations. The observed morphologies were categorized into several types, and seasonal changes in the occurrence rates of each type with seasonal variations were calculated. Preliminary tank experiments were conducted from September to November 2023 to investigate the relationship between the morphology of burrow

openings and the behavior of *U. major* within the burrows. Three tanks measuring 100 cm in length, 10 cm in width, and 130 cm in height were used (Fig 5a). Each tank was filled with natural sediment from Mangoku-ura, artificial seawater replicating the local water temperature and chlorophyll concentration, and one individual *U. major* collected from Mangoku-ura. The *U. major* used in the experiment were two male and one female adult specimens. To simulate tidal fluctuations, seawater was carefully removed from each tank twice daily using a 50 mL pipette. The tanks were then maintained in a low-tide state for approximately 2.5 hours before slowly reintroducing fresh seawater. The process of burrow formation and the behavior within the burrow were observed through the cross-section of the tank.

Additionally, live specimens were collected. Living specimens were collected by targeting burrows within quadrats using a writing brush and yabby pump (Poseidon, Aichi, Japan). The collected specimens were immediately sexed onsite, and upon returning to the laboratory, the carapace length (CL) was measured using an electronic caliper.

Water temperature and chlorophyll concentration were measured at the study sites. Water temperature measurements were conducted on the surface seawater around the survey sites and within the tidal pools using a water thermometer (CM-31P, TOA-DKK, Tokyo, Japan). The chlorophyll concentration in seawater (μg/μL) was measured as an indicator of food quantity. The concentration was measured using a chlorophyll sensor (CHL-30N; Kasahara Chemical Instruments Corp., Saitama, Japan).

The survey periods for Lake Akkeshi were July 2022 and June and October 2023. For the other study sites, investigations were conducted between March and November from 2021 to 2023. All surveys were conducted during the low-tide period of spring tides. Additionally, nighttime surveys were conducted in November because of the occurrence of nocturnal tides. In this study, March through May was designated as spring, June through July as summer, September through October as autumn, and November as winter.

## Collection of sediment samples for sedDNA analysis

Sediment samples from Lake Akkeshi were collected in June and October of 2023 (Table 2). Lake Akkeshi experiences seasonal changes in factors such as air and water temperatures approximately one month later than the other survey sites. Consequently, the survey results for each month could be compared with those from May and September at other survey

**Table 2. Sediment sample collection schedule for sedDNA analysis in Lakes Akkeshi, Mangoku-ura, and Sanbanze. The recorded temperatures near the study sites during the survey period were obtained from the Japan Meteorological Agency (https://www.data.jma.go.jp/stats/etrn/index.php).**

| Collected location | Collected date | Weather conditions | Time intervals | Atmospheric temperature (Maximum) |
|---|---|---|---|---|
| Lake Akkeshi | 19-Jun-23 | Sunny | 9:30 AM to 11:30 AM | 20.4°C |
| | 12-Oct-23 | Sunny | 8:00 AM to 11:00 AM | 18.5°C |
| Mangoku-ura | 21-Apr-23 | Sunny | 9:00 AM to 12:00 PM | 18.2°C |
| | 19-May-23 | Cloudy | 9:00 AM to 1:00 PM | 15.5°C |
| | 31-Jul-23 | Sunny | 10:00 AM to 1:00 PM | 34.4°C |
| | 14-Sep-23 | Cloudy | 9:00 AM to 11:30 PM | 32.5°C |
| | 14-Nov-23 | Sunny | 7:00 PM to 9:30 PM | 11.2°C |
| Sanbanze | 25-Apr-23 | Cloudy | 10:00 AM to 2:30 PM | 13.1°C |
| | 23-May-23 | Cloudy | 10:00 AM to 2:00 PM | 15.8°C |
| | 21-Jul-23 | Sunny | 10:30 AM to 2:00 PM | 32.6°C |
| | 28-Sep-23 | Cloudy | 9:30 AM to 2:00 PM | 28.5°C |
| | 27-Nov-23 | Sunny | 9:00 PM to 11:30 PM | 7.4°C |

sites. Since there is no burrow-rich area at this site, three 80 m transects (A, B, and C) were established, with approximately 1 g of surface sediment collected at 20 m intervals along each transect (Fig 2a). At Mangoku-ura and Sanbanze, sediment sampling was conducted in April, May, July, September, and November of 2023. Approximately 1 g of surface sediment was randomly collected from six sites within burrow-rich areas. As samples outside the burrow-rich areas, surface sediments were collected at 20 m intervals within a radius of 40 to 100 m, extending in four different directions (Directions A, B, C, and D) from the initial sites within the burrow-rich areas (Figs 2b and d). Moreover, owing to the constraints applicable to both study sites in April, sampling was conducted only in A-transect and B.

A plastic spoon sterilized with chlorine bleach was used for sampling. A new sterilized spoon was used for each sampling. The collected sediment samples were frozen using a White Freezer (Yuaikasei, Hyogo, Japan), transported to the laboratory in a frozen state, and stored at −20 °C until the sedDNA extraction. Additionally, visual observations of the activity of marine benthos, including the species inhabiting the study area, were made.

## sedDNA extraction and quantification of the total sedDNA concentration

The sediment samples were transferred to a mortar while kept in a frozen state and homogenized. Subsequently, 0.25 g of the homogenized samples were used to extract total sedDNA using a DNeasy PowerSoil Pro Kit (Qiagen, Hilden, Germany). The extracted sedDNA was stored at −20 °C until further analysis. The total concentration of the sedDNA was determined using a fluorescence-based method to confirm successful DNA extraction. This measurement was conducted using the QuantiFluor dsDNA System kit (Promega Corporation, Madison, WI, USA) as the fluorescence reagent and a Quantus fluorometer (Promega, Walldorf, Germany) as the fluorescence measurement instrument (S1 Table in S1 File).

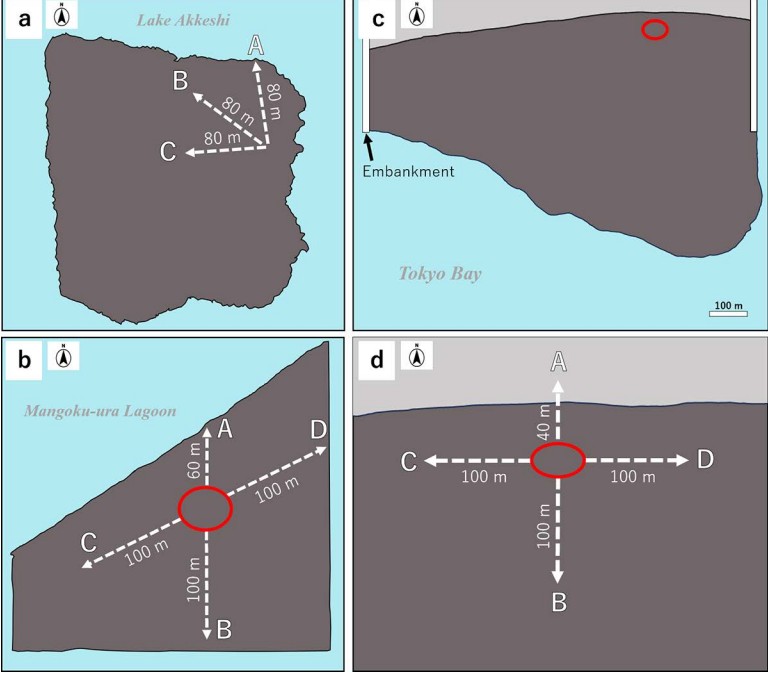

**Fig 2. A top-down view of sediment sampling sites in each study site.** (a) Lake Akkeshi, (b) Mangoku-ura, (c) Sanbanze. (d): Enlarged image of c. Red circles indicate the burrow-rich areas. The gray areas represent the upper intertidal zone, while the brown areas indicate tidal flats.

## Species-specific primers and probe for *U. major*

Through molecular phylogenetic analysis targeting the cytochrome *c* oxidase I (COI) region of the mitochondrial DNA of *U. major* conducted by [15], this species was genetically divided into four groups, with individuals from Japan present in all groups. Therefore, it is desirable to design a primer-probe set capable of detecting individuals from all groups. Thus, a region with nucleotide sequences common to all groups and distinct from the closely related species *U. yokoyai* was explored and designated as the sequence for the species-specific primer-probe for *U. major*. The primer (forward primer: UMFW2208, Reverse primer: UMRV2208) and probe (TaqMan MGB Probe: UMPb1) sequences are listed in Table 3. The PCR amplification size achieved through PCR was 94 bp.

In order to validate the species-specificity of the designed primer-probe set, quantitative polymerase chain reactions (qPCR) were performed on genomic DNA extracted from the muscle tissues of *U. major*, its closely related species *U. yokoyai*, and the dominant species at each survey site, including Asari clam *R. philippinarum*, hermit crab *Pagurus minutus* and horn snail *Batillaria attramentaria* (refer to next paragraph for the composition of the qPCR reaction solution and PCR conditions). Only the DNA of *U. major* was amplified, substantiating its species specificity (S1 Fig).

## Quantification of sedDNA concentration using qPCR

The qPCR used in this study was performed using the StepOne Real-Time PCR system (Thermo Fisher Scientific, Waltham, MA, USA). The total volume of the qPCR reaction mixture was 20 μL, which included 2 μL of DNA extraction solution, 10 μL of TaqMan

**Table 3. Sequence alignment of *U. major* (Group 1 to 4) and *U. yokoyai*. The target region was a COI fragment using the following primers and probe: a: Forward primer (UMFW2208), b: Reverse primer (UMRV2208), and c: TaqMan Probe (UMPb1).**

| a | Species | Nucleotide sequences |
|---|---------|----------------------|
| | *U. major* (Group 1) | ···· TAAGAGGAATAGTAGAAAGAGGTGTTGGAACAGG···· |
| | *U. major* (Group 2) | ···· TAAGAGGAATAGTAGAAAGAGGTGTTGGGACAGG···· |
| | *U. major* (Group 3) | ···· TAAGAGGAATAGTAGAAAGAGGTGTTGGGACAGG···· |
| | *U. major* (Group 4) | ···· TAAGAGGAATAGTAGAAAGAGGTGTTGGGACAGG···· |
| | *U. yokoyai* | ···· TAAGAGGAATAGTAGAGAGTGGTGTAGGAACAGG···· |
| | UMFW2208 | ······ GGAATAGTAGAAAGAGGTGTTGG |
| b | Species | Nucleotide sequences |
| | *U. major* (Group 1) | ···· CGCCGGTGCTTCCGTTGACATAGGTATTTTC···· |
| | *U. major* (Group 2) | ···· CGCCGGTGCTTCCGTTGACATGGGTATTTTT···· |
| | *U. major* (Group 3) | ···· CGCCGGTGCTTCCGTTGACATGGGTATTTTT···· |
| | *U. major* (Group 4) | ···· CGCAGGTGCTTCCGTTGATATGGGTATTTTT···· |
| | *U. yokoyai* | ···· TGCTGGTGCTTCTGTTGATATAGGAATTTTT···· |
| | UMRV2208 | ······ GTGCTTCCGTTGACATAGGTA |
| c | Species | Nucleotide sequences |
| | *U. major* (Group 1) | ···· CCTCCACTATCAGCTGCTATTGCCCACGCCGGTG···· |
| | *U. major* (Group 2) | ···· CCCCCACTATCAGCTGCCATTGCCCACGCCGGTG···· |
| | *U. major* (Group 3) | ···· CCTCCACTATCAGCTGCTATTGCCCACGCCGGTG···· |
| | *U. major* (Group 4) | ···· CCTCCACTATCAGCTGCTATTGCCCACGCCGGTG···· |
| | *U. yokoyai* | ···· CCTCCCCTATCCGCAGCTATCGCCCATGCTGGTG···· |
| | UMPb1 | ······ ACTATCAGCTGCTATTGCCCACGC |

Fast Advanced Master Mix (Thermo Fisher Scientific, Waltham, MA, USA), 1.8 uL each of 10 pmol/uL primers (final concentration 900 nM), 0.5 μL of 10 pmol/μL TaqMan MGB probe (250 nM), and 3.9 μL of nuclease-free water. qPCR was performed using a dilution series ranging from $3 \times 10^6$ to $10^1$ of artificially synthesized DNA based on the COI gene sequence of *U. major*. The positive control consisted of a solution in which the genomic DNA of this species was diluted to 1/1000, and nuclease-free water was used as the negative control. qPCR was conducted with the following thermal cycling profile: 20 s at 95 °C, followed by 40 cycles of denaturing at 95 °C for 1 s and annealing/extension at 55 °C for 20 s. The optimization of annealing temperature was determined by testing three temperatures (60 °C, 58 °C, and 55 °C) using sedDNA extracted from surface sediment collected in the Mangoku-ura in April 2023, and the annealing temperature that yielded the highest number of detections of *U. major* derived sedDNA, i.e., 55 °C, was used (S2 Table in S1 File). For unknown samples, three measurements were conducted for each sample, and the highest concentration (number of copies) was used for data analysis. The data acceptance criteria were based on [41], with the slope of the standard curve ranging from $-3.428$ to $-4.077$, $R^2$ values of 0.976 or higher, and PCR efficiency in the range of 75.908 to 95.741%. Furthermore, the data unit for the concentration was designated as copies/g sediment. All experiments were conducted in a clean room.

## Data analysis

The Tukey-Kramer test (Q Value ($\alpha = 0.05$)) was used for data analysis. For the samples from Lake Akkeshi, it was examined whether seasonal differences in sedDNA concentrations occurred between the transects. For Mangoku-ura and Sanbanze, it was examined whether there were significant seasonal differences in sedDNA concentrations between the burrow-rich areas and other areas, within the burrow-rich areas, and across the different transects. However, if sedDNA was not detected at all in either the burrow-rich area or the other areas, tests were not conducted for those cases. When data below the detection limit were included, the concentration values were tested using the detection limit value. The detection limit value (x) was calculated by transforming the equation of the calibration curve obtained from a linear function as follows, with x initially obtained in copies/μL and subsequently converted to copies/g sediment:

$$x = (41 - b)/a$$

The value of 41 represents the assumed threshold Ct value, where a is the slope and b is the Y-intercept. The values for the slope and Y-intercept used were those listed in S4 Table in S1 File.

## Results

### Seasonal variations in the number of burrow openings

The distribution of *U. major* burrows was confirmed at all study sites (Fig 3a). The mean number of burrow openings within the quadrat was highest in Lake Akkeshi, showing relatively consistent numbers of approximately 50 burrow openings maintained from June to October without significant fluctuations. In other words, it was estimated that approximately 25 individuals of *U. major* inhabited the $50 \times 50$ cm quadrant in Lake Akkeshi. There was a tendency for an increase in the number of burrow openings in Mangoku-ura and Sanbanze when the water temperature and food availability started to rise, particularly from March to May (Figs 3a and f). The estimated abundance was significantly lower than that in Lake Akkeshi, with a maximum of approximately three individuals at each study site. They exhibited a decreasing trend with seasonal changes, and a significant

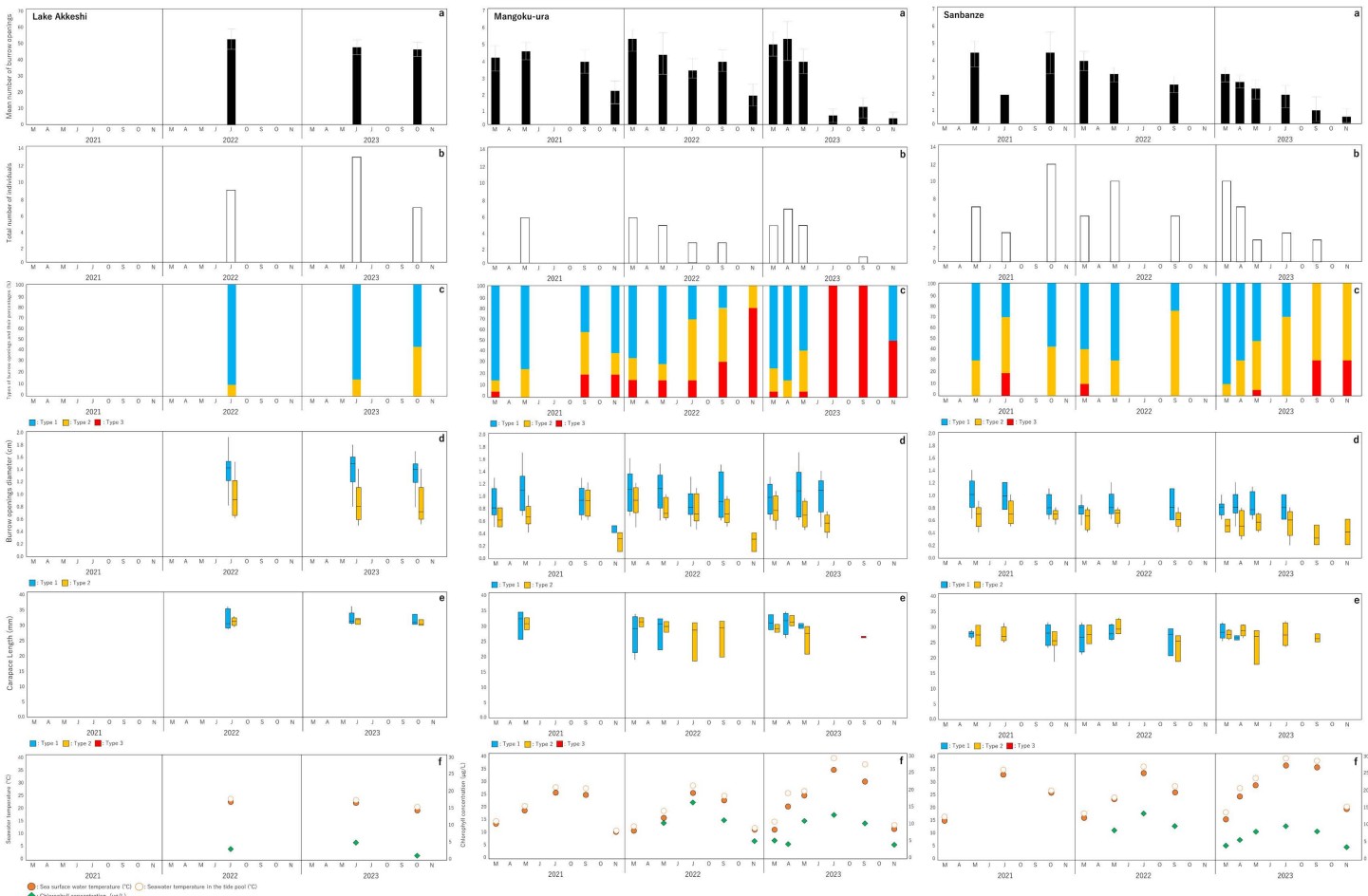

**Fig 3. The mean number of burrows (a), total number of individuals (b), types of burrow openings and their percentages (c), relationship between the morphological types and diameter of burrow openings (d), relationship between the types and CL size (e), and seawater temperature and chlorophyll concentration (f) at each study sites.** Blank areas indicate months where no survey was conducted or where data was not obtained.

reduction was observed at many study sites in November when both the water temperature and chlorophyll concentration decreased. There was a significant decrease in the number of burrow openings from summer to autumn, i.e., when seawater temperatures increase. Mangoku-ura and Sanbanze showed a substantial decrease in the number of burrow openings in July and September 2023. Notably, only three burrows were discovered in Mangoku-ura in July 2023, and several quadrats showed no burrows (S3 Table in S1 File). The variation in the total number of collected individuals revealed a seasonal pattern akin to that of the burrow distribution density (Fig 3b). Many egg-bearing females were collected at Lake Akkeshi in July, at Mangoku-ura in March and April, and at Sanbanze in March (Table 4). November, which was characterized by a decrease in the number of burrow openings, did not yield any specimens.

## Seasonal changes in the morphology of burrow openings

The morphology of burrow openings can be broadly classified into three discernible types, as shown in Fig 4. Type 1 is characterized by an unadorned circular configuration (Fig 4a),

**Table 4. The number of *U. major* individuals according to sex was determined at each study site. Parentheses indicate the number of egg-bearing individuals.**

| Number of individuals by sex (male: female) | 2021 | | | | | | 2022 | | | | | 2023 | | | | | | | |
|---|---|---|---|---|---|---|---|---|---|---|---|---|---|---|---|---|---|---|---|
| | March | May | July | September | Octover | November | March | May | July | September | November | March | April | May | June | July | September | Octover | November |
| Lake Akkeshi | | | | | | | | | 2: 7 (3) | | | | | | 8: 5 | | | 5: 2 | |
| Mangoku-ura | | 4: 2 | | | | | 1: 5 (3) | 2: 3 | 3: 0 | 2: 1 | | 2: 3 (2) | 1: 6 (6) | 2: 1 | | | 1: 0 | | |
| Sanbanze | | 3: 4 | 3: 1 | | 8: 4 | | 1: 5 (2) | 6: 4 | | 3: 3 | | 2: 8 (3) | 5: 2 | 0: 3 | | 3: 1 | 2: 1 | | |

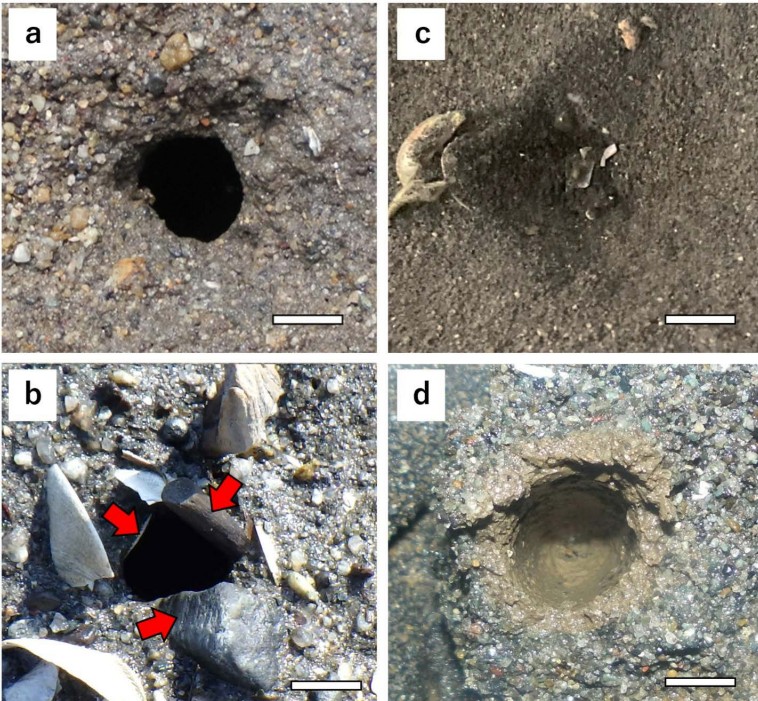

**Fig 4. Morphology of *U. major* burrow openings.** (a) Type 1, (b) Type 2, red arrows indicate the pebbles and shells of the Asari clam *R. philippinarum*. (c) Type 3, (d) Burrow openings, as observed in the interior of c. The oxidized yellow-brown portion represents the burrow cross-section. The scale bar represents 1 cm.

Type 2 by the circumferential arrangement of pebbles or shell fragments (Fig 4b), and Type 3 by burrows blocked by sediment or shell fragments, resulting in a significantly reduced diameter of the openings (Fig 4c). Visually determining the position of Type 3 burrows can be challenging; however, a slightly concave feature at the openings serves as a distinguishing marker. Typically, *U. major* burrows extend almost vertically downward from the openings to depths ranging from 6.5 to 52.7 cm [9]. A stick approximately 10 cm in length would usually enter the burrow completely if it were a burrow of this species. By pressing the writing brushes against the depressed area, we determined whether it was a burrow. Additionally, diameter measurements were not conducted because it was not possible to determine the outer edge of the opening.

In the case of Lake Akkeshi, long-term variations remain uncertain owing to the limited number of months surveyed. Type 1 was predominant, with occasional sightings of Type 2 (Fig 3c). Notably, the longest diameter of the burrow openings reached a maximum of 1.9 cm, making it the largest size observed at the study site (Fig 3d). All types of burrow openings were observed in Mangoku-ura and Sanbanze. From March to May, when the number of burrow openings was high, Type 1 was dominant in spring. However, from July to September, when the number of burrows showed a decreasing trend, the proportions of Types 2 and 3 increased. Additionally, in July and September 2023, when the number of burrow openings significantly decreased in Mangoku-ura, the proportion of Type 3 increased substantially (Fig 3c). By November, almost all cases were Type 3. The longest diameter of burrow openings was approximately 1.7 cm at maximum during spring when Type 1 was prevalent. However, in winter, when Type 3 burrows were more common, the burrow diameter became so small that it was difficult to locate them. In other words, during Type 1, the size of the burrow openings

reaches its maximum, making the burrows easier to detect. In contrast, when the proportion of Type 3 increases, the burrows become harder to detect, resulting in an apparent decrease in the number of burrows. Both males and females, ranging from subadults to adults with a CL of 25 mm or less, were collected from burrows exhibiting Types 1 and 2 (Fig 3e). Types 1 and 2 seem to be formed irrespective of sex or growth stage. On the other hand, the only individual collected from Type 3 burrows was a male adult collected in Mangoku-ura in September 2023. However, as revealed by the tank experiments, female adults were also found to construct Type 3 burrows (Table 5). Type 3 burrows appear to be formed regardless of sex.

In the tank experiment, between September and October, when the water temperature was maintained at 15°C to 20°C, all individuals kept Type 1 burrows (Fig 5b). However, when the water temperature dropped to around 12 °C in November, the burrows became blocked with sediment and shell fragments, resulting in the formation of Type 3 burrow openings (Fig 5c). The transition from Type 1 to Type 3 occurred within one day, and the experimental individuals modified their burrow openings regardless of tidal conditions. By altering the shape of the burrow openings, they seemed to respond immediately to changes in water temperature. Only

**Table 5. Characteristics of the individuals used in the tank experiment and types of burrow openings morphology formed each month.**

| Sample code | Carapace Length (mm) | Sex | Type of burrow openings morphology | | |
|---|---|---|---|---|---|
| | | | September | October | November |
| UM-1 | 25.7 | Male | Type 1 | Type 1 | Type 3 |
| UM-2 | 28.5 | Female | Type 1 | Type 1 | Type 3 |
| UM-3 | 27.2 | Male | Type 1 | Type 1 | Type 3 |

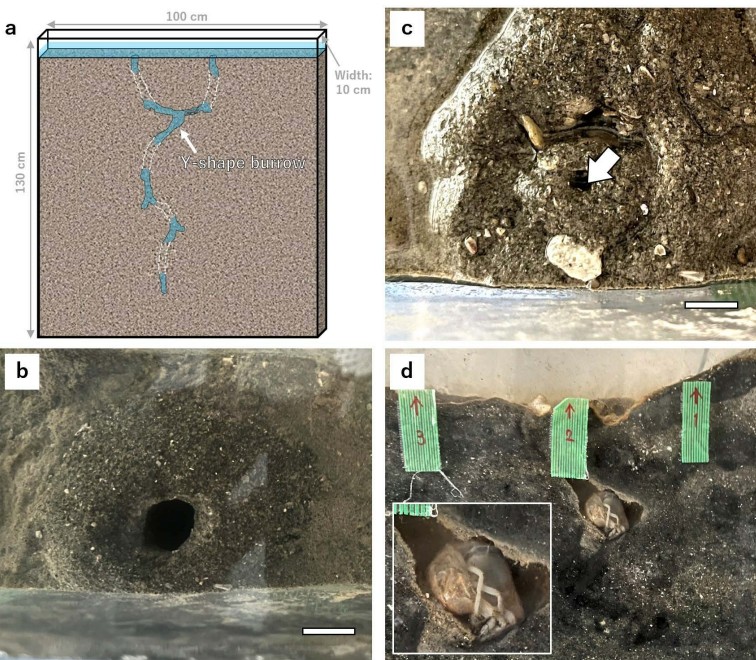

**Fig 5. Simple tank experiment with *U. major* using slim tanks.** Burrow cross-sections were observed intermittently. (a) Model of tank used in experiment, (b) Type 1 burrow opening formed between September and October 2023, (c) Type 3 burrow opening formed in November 2023. White arrows indicate openings, (d) The reared individual (UM-3) curled and barely moved inside the burrow in November. Scale bar represents 1 cm.

one individual (UM-3) was observed for behavior through the cross-section of the burrow. UM-3 formed a Type 3 burrow opening and then curled up, remaining mostly inactive within the burrow (Fig 5d). This behavior may indicate hibernation. At least in adult *U. major*, it appears that they block their burrows and form Type 3 burrows for hibernation when water temperatures drop during winter.

## Seasonal changes in sedDNA concentration

In Lake Akkeshi, where burrows are widely and densely distributed, a notably high number of detections were made in June, revealing uniformly high concentrations of sedDNA ranging from $10^4$ to $10^5$ copies/g sediment across all transects (Fig 6). However, a considerable decrease in the number of detections was observed during October. Furthermore, the results of the Tukey–Kramer test (Q Value ($\alpha = 0.05$)) showed that the samples from A-transect in October were significantly lower in concentration compared to the samples from June (S7 Table in S1 File).

The investigation in Mangoku-ura was limited to A and B-transects in April; however, a high concentration of sedDNA, ranging from $10^4$ to $10^5$ copies/g sediment, was detected across a wide range (Fig 6). Within the burrow-rich area, it was detected at all six sites. Furthermore, the results of the Tukey–Kramer test (Q Value ($\alpha = 0.05$)) revealed no significant difference in concentration between the burrow-rich area and other regions (Fig 7 and S6 Table in S1 File), indicating a widespread distribution of sedDNA with similar concentrations. By contrast, the number of detections was low in areas without burrows in May. Significant differences in concentration were observed between the burrow-rich area and other areas (Fig 7 and S6 Table in S1 File), confirming the concentration of high sedDNA concentration

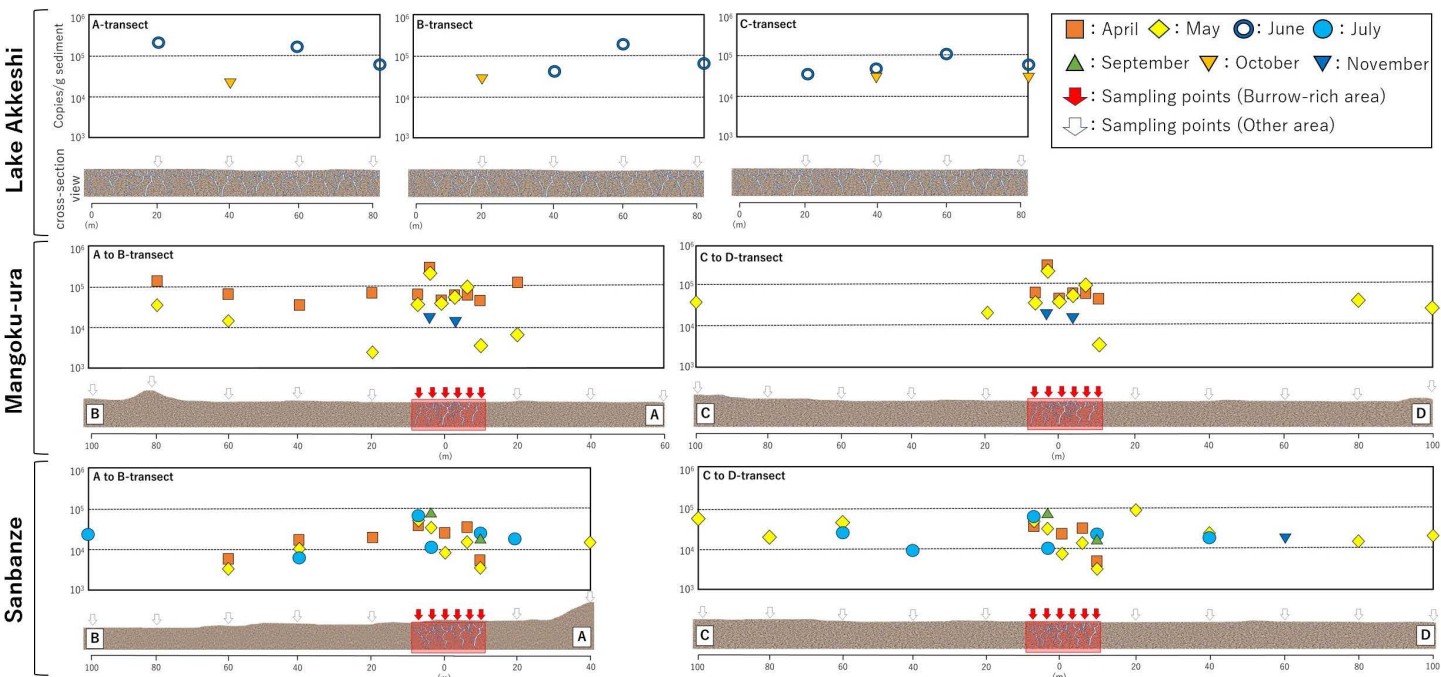

**Fig 6. The distribution of *U. major* burrows and sedDNA concentrations at the survey sites.** (a) Lake Akkeshi, (b) Mangoku-ura, (c) Sanbanze. Red squares indicate Burow-rich areas. The blank spaces indicate where concentration data could not be obtained. The irregularities on the seabed reflect the results of on-site topographical observations. The depth of the seabed cross-section and the shape of *U. major* burrows are conceptual images.

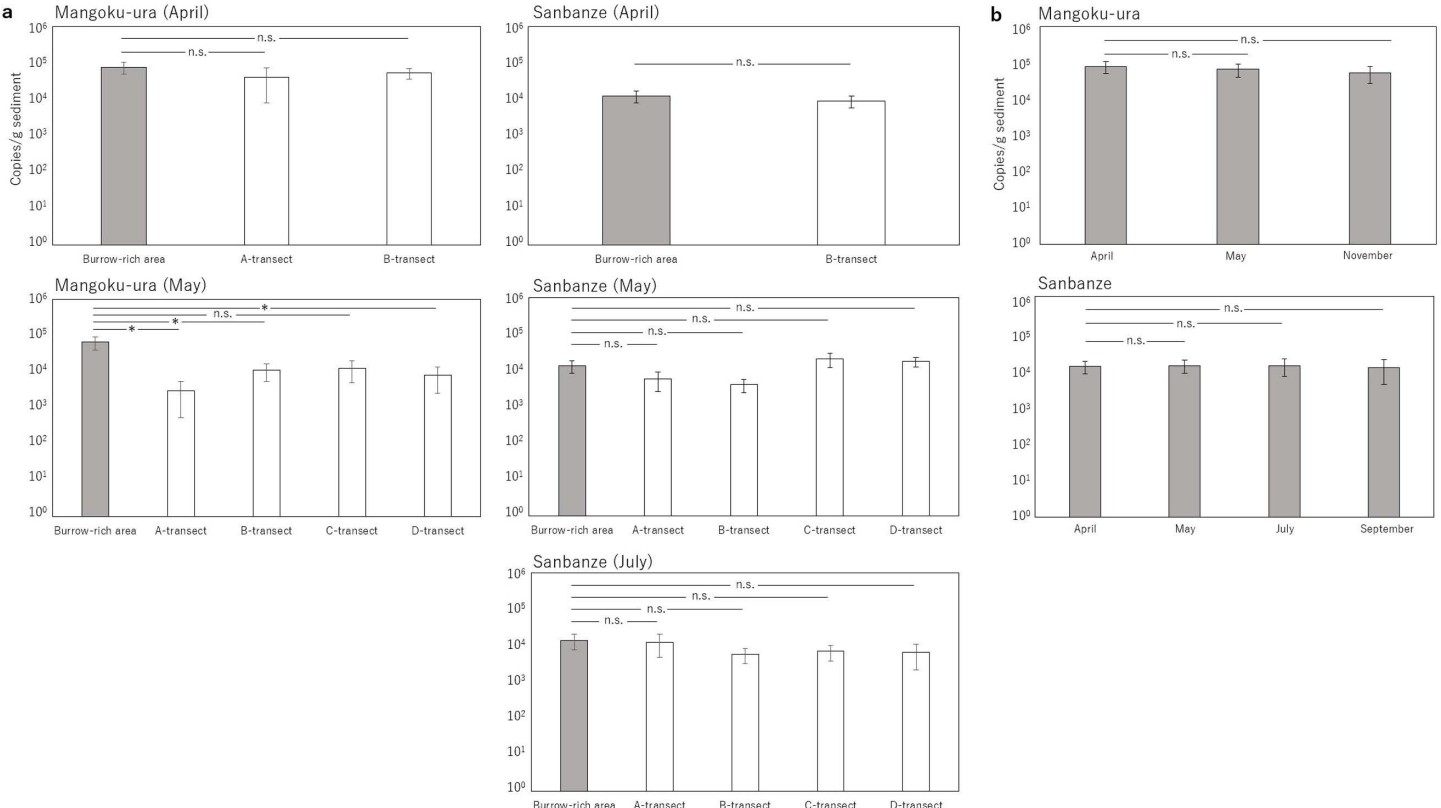

**Fig 7. Mean sedDNA concentrations and their significance between the burrow-rich area and other areas (A to D transects) (a), as well as within the burrow-rich area (b).** However, Tukey–Kramer test (Q Value (α = 0.05)) was not conducted if sedDNA was not detected in either the burrow-rich area or other areas. *: significant difference, n.s.: not significant.

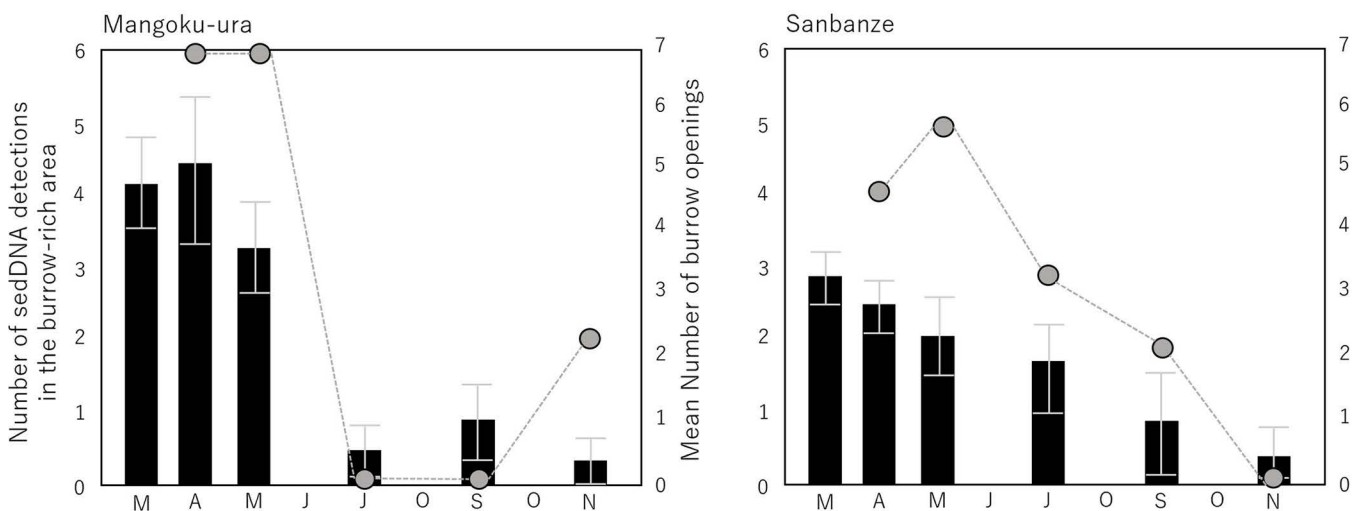

**Fig 8. Relationship between the number of burrow openings (bar graph) and the number of sedDNA detections (line graph) in burrow-rich areas.**

within the burrow-rich area. However, from July onwards, when the number of burrow openings decreased, the number of sedDNA detections significantly declined in all subsequent months (Fig 8). In July and September, although the total sedDNA concentration was measurable (S1 Table in S1 File), no sedDNA from *U. major* was detected. In November, sedDNA was detected only within the burrow-rich area, with the number of detections limited to two. The sedDNA concentration within the burrow-rich area showed variability in the number of detections, but no significant seasonal differences were observed (Fig 7b). Additionally, when testing the difference in sedDNA concentrations month by month, it was found that the B-transect in April, where sedDNA was detected extensively, exhibited significantly higher concentrations compared to other months (S7 Table in S1 File).

In Sanbanze, sedDNA concentrations ranging from $10^3$ to $10^5$ copies/g sediment were detected from April to July. No significant differences in concentrations were observed between the burrow-rich area and the other areas in any month (Fig 7 and S7 Table in S1 File). Furthermore, there was no significant difference in the concentration of sedDNA between months (S7 Table in S1 File). In May, sedDNA was detected at both ends of C and D-transects, 100 m from the Burrow-rich area. On the other hand, in July, the detected range was extensive, but the number of detections exhibited a slight decline, with the burrow-rich area being detected in only three out of six sites (Fig 8). After the decrease in the number of burrow openings from September onwards, the number of sedDNA detections further declined (Fig 8). In September, detections were made at two locations within the burrow-rich area, and in November, only one location along the D-transect showed detections.

In addition, there were two representative pieces of evidence for the visually observable activity of *U. major*. One notable phenomenon is sand volcanoes [47]. This phenomenon is frequently observed when a species excavates its burrows or engages in maintenance activities (S1 Appendix). Seawater containing sediment particles emanated from the burrow. The second phenomenon is presumed to occur during feeding and is characterized by the rhythmic spouting of seawater exclusively from within the burrow (S2 Appendix). Upogebiid shrimp are believed to consume seawater from one burrow opening, engage in filter feeding, and expel seawater from another burrow opening [18]. It was presumed that the observed expulsion of seawater resulted from this process. These two phenomena indicate the release of water and sediment from the burrow to the exterior environment. This may have led to the release of eDNA or sedDNA produced by *U. major* within the burrows onto the seafloor. In other words, these phenomena are likely to significantly affect the concentration and number of detected sedDNA originating from this species. Sand volcanoes and seawater spouting were frequently observed in June at Lake Akkeshi, from April to May in Mangoku-ura, and from April to July at Sanbanze. These phenomena were rarely observed at any of the surveyed sites in the other months.

## Discussion

### Correlation between sedDNA concentration and the abundance of *U. major*

In Lake Akkeshi, where burrows are distributed over a wide area, sedDNA of similar concentrations was detected from a wide area in June. Additionally, in Mangoku-ura in May, when a relatively high number of burrows were observed, the sedDNA detected in burrow-rich areas was significantly higher in concentration than in other areas. In other words, in both cases, a correlation between the abundance of *U. major* and sedDNA concentration was observed. What is common to both cases is the topography and the timing.

Lake Akkeshi and Mangoku-ura are both situated in bag-shaped lagoons, where the weak seawater flow results in minimal fluctuations in the bottom environment. Ripple marks and similar features are rarely observed on the seabed of tidal flats due to the minimal movement

of bottom sediments. Additionally, although river water flow can potentially cause sediment displacement, Mangoku-ura lacks significant river inflows, making such an influence unlikely. In the case of Lake Akkeshi, several small rivers flow into the lake, with the largest one entering from the north. The seawater flow within Lake Akkeshi differs between the northern and southern regions; during high tide, the water in the northern part flows counterclockwise, while in the southern part, it flows clockwise [48]. This pattern is reversed during low tides. As the survey site is located on the southern side of the lake, seawater flow differs from that on the northern side, and the influence of river water flow is expected to be minimal. Therefore, it is considered that Mangoku-ura and Lake Akkeshi are environments where sedDNA deposited on the seabed is less likely to advect to distant locations.

On the other hand, it was assumed that the months during which the surveys were conducted at both sites coincided with the growth period of this species. The life cycle of crustaceans includes periods of high activity, such as spawning and growth periods, and periods of low activity, such as hibernation. In Mangoku-ura, seven individuals of *U. major* were collected in April, six of whom were egg-bearing females. These egg-bearing individuals were transported to the laboratory and kept in buckets, where all six individuals laid their eggs in a short span of seven days, ultimately resulting in the hatching of a substantial number of zoea larvae. According to [49], the spawning period of *U. major* in Tokyo Bay spans from December to May. There is no existing knowledge regarding the spawning period of this species in the Tohoku region, where the Mangoku-ura is located. However, based on the results of this study, there is a high likelihood that the peak spawning will occur in April. The reason for the higher number of females collected in April remains uncertain. To ensure the successful release of larvae from deep within the burrows, egg-bearing individuals may have been located in the upper part of the burrows, which might have been one of the factors making them easier to collect. While, despite June being considered the peak spawning period in Hokkaido [50], in Lake Akkeshi, egg-bearing individuals were not collected in months other than July. In Lake Akkeshi, there is a possibility that the peak occurs slightly later around July. At both study sites, only normal (non-egg-bearing) individuals were collected in May and June, and the activity of this species was high, as evidenced by the frequent observations of sand volcanoes and seawater spouting. Therefore, it is highly likely that May in Mangoku-ura represents the post-peak spawning growth period, while June in Lake Akkeshi represents the pre-peak spawning growth period. Additionally, attention should be paid to the morphology of burrow openings observed during this period. At both study sites, Type 1 burrows were frequently observed. The relationship between the morphology of burrow openings and the physiological activity of *U. major* has not been sufficiently studied. However, Type 1 burrows may be observed during periods of high activity for this species, such as spawning or growth period. Type 1 burrows, characterized by their larger diameter, likely facilitate the substantial deposition of sediment particles and seawater containing eDNA from *U. major* onto the seafloor through sand volcano and seawater spouting. Although it is assumed that a significant portion of the produced sedDNA accumulates around the burrows, the absence of strong water currents suggests that it remained in place.

Based on the above, it appears that in a bag-shaped lagoon with minimal changes in sedimentary environments, a correlation between abundance and sedDNA concentration can be observed during the growth period of this species.

## Potential causes of fluctuations in sedDNA concentration; Life cycles and topographical differences

In Mangoku-ura, the concentration of the detected sedDNA was significantly higher in April than that in other months. There was no significant difference in the concentration between

the burrow-rich area and other areas, suggesting a widespread distribution of similar concentrations. In other words, there was no correlation between the abundance of *U. major* and sedDNA concentration during this month. This could have involved the spawning period. In Mangoku-ura, this month corresponds to the peak spawning period; hence, it is highly likely that a large number of larvae, egg remnants, and mucus from spawning events were discharged. These serve as sources of sedDNA, potentially influencing their concentrations and detection ranges. While it is an example from fish species, it has been reported that *Acanthopagrus schlegelii* exhibits an increase in both eDNA concentration and detection range during its spawning period [51]. In the survey conducted in April, the larvae of *U. major* were not observed. However, their dispersal over a wide area during the tidal cycle may have contributed to the increase in sedDNA concentration and expansion of the detection range.

While, because egg-bearing individuals were collected only in March in Sanbanze, it was assumed that there were many individuals during the growth period in April and May. However, no correlation was observed between the sedDNA concentration and abundance. There is a possibility that strong waves are involved in this phenomenon. Sanbanze, a foreshore tidal flat characterized by waves measuring approximately 0.5 to 1 m in height year-round, is marked by the intense displacement of surface sediments due to the action of swash and backwash. Although this example specifically pertains to aquatic eDNA, when confronted with robust water currents, there exists a proclivity for eDNA to be transported over extended distances while preserving its elevated concentrations, as elucidated by [52]. Consequently, it can be surmised that the pronounced wave current at Sanbanze facilitated the transport of sedDNA with high concentrations to remote areas, consequently mitigating the disparities in concentration between the burrow-rich area and other environs, potentially enlarged the scope of detection. Furthermore, it is imperative to consider the anthropogenic ramifications. Sanbanze is a locale frequented by individuals daily, and numerous people traverse the intertidal expanse. Additionally, a considerable portion of the intertidal territory lacks codified fishing rights; as a result, visitors excavate the bottom sediment to collect mollusks and crustaceans. Consequently, it cannot be ruled out that sedDNA may have been widely dispersed across the area due to anthropogenic disturbances.

On the other hand, during the winter season (November), when the water temperature and chlorophyll concentration decreased, the number of sedDNA detected was very low at all survey sites. Additionally, it was noteworthy that the morphology of most burrow openings was Type 3. These factors may be associated with decreased activity (hibernation) during winter. There has been no previous research on the hibernation of *U. major*, so the specific conditions that trigger hibernation are unknown. However, *Upogebia deltaura*, which inhabits the Mediterranean and eastern Atlantic, is believed to hibernate in the deep recesses of burrows during winter [53]. Similarly, ghost crabs *Ocypode cordimanus*, which also dig deep burrows in sandy beaches, exhibit a behavior of sealing the burrow openings with sediment during hibernation [54]. In the tank experiment conducted in this study, Type 3 burrows were formed in November, and the experimental individuals exhibited behavior suggesting hibernation, as they curled up and became inactive. In other words, in November, the reduced physiological activity of the species due to hibernation or preparation for hibernation may have led to a decrease in sedDNA production, resulting in a lower number of detections. The white-clawed crayfish *Autropotamobius pallipes*, which belong to the same order as *U. major*, are known to exhibit a decrease in the number of eDNA detections during the winter months when their activity declines [55]. Furthermore, the brackish-water clam *Corbicula japonica* enters dormancy in the bottom sediment during the winter months, leading to a decrease in eDNA concentration during that period [56]. The results of this study are consistent with those of previous studies.

## Changes in physiological activity due to environmental fluctuations affect the concentration of sedDNA

From summer to autumn (July to October), the concentration and number of sedDNA detected significantly decreased at each study site. Specifically, in Mangoku-ura, the number of burrows drastically decreased compared to previous years, and sedDNA was not detected. In Lake Akkeshi, the number of detections decreased, despite the presence of numerous burrows. The prominent factor contributing to this appears to be the "abnormal heat." From June to October 2023, Japan experienced a prolonged heat wave, resulting in the highest average summer temperatures recorded since statistical records began in 1898 [57]. In the Tohoku region, the average temperature surpassed the normal level for three consecutive months from June to August, and a marine heat wave was observed [58]. In Mangoku-ura, between July and September, although the chlorophyll concentration remained typical at all survey sites, the sea surface temperatures persisted in the range of 30 °C, escalating to around 40 °C in tide pools, marking the highest temperatures observed in the past two years (Fig 3f). Additionally, in October, the seawater temperature in eastern Hokkaido was more than 2 °C higher than the previous year, marking the highest in the past 30 years [59]. The seawater temperature in Lake Akkeshi during October measured around 18 °C, surpassing 20 °C within the tidal pools (Fig 3f). Heat stress affects the activity and physiological processes of aquatic poikilothermic animals, including crustaceans [60, 61]. Although the optimal and lethal temperatures for *U. major* have not been precisely determined, it is suggested that during the summer season, they dig deeper burrows to escape high temperatures [17]. Additionally, the congeneric species endemic to South Africa, *Upogebia africana*, the lethal temperature is established at 29 °C, and it is known that when the water temperature exceeds 25 °C, they seal their burrow openings with mud [62]. At all study sites, there was a significant increase in Type 2 or 3 burrows from summer to autumn, narrowing the burrow openings. Hence, it is unlikely that this species maintains high performance under such conditions. The increase in seawater temperature may have led to a decrease in the physiological activity of *U. major*, resulting in a potential reduction in sedDNA concentration and the number of detections.

Additionally, in Mangoku-ura, ground uplift may also be involved. In recent years, ground uplift caused by aftershocks of the Great East Japan Earthquake in 2011 has been observed in Mangoku-ura [63]. On the nearby Oshika Peninsula, located a short distance from Mangoku-ura, there has been a recorded uplift of 50 cm per decade since the earthquake [44]. This uplift extended the duration of low tide in the tidal flat. During the spring tides in July and September, when the investigation was conducted, the tidal flat remained exposed for over five hours. Both months witnessed a large number of *R. philippinarum* deaths in tidal flats (Okoshi, unpublished data). Although no carcasses of *U. major* were found, it cannot be ruled out that the cumulative effects of prolonged exposure and high water temperatures were involved in the variation in sedDNA concentration.

Another interpretation is that sedDNA was rapidly degraded during these periods. Various factors contribute to eDNA degradation, including UV radiation, bacteria, and DNA-degrading enzymes [64, 65]. Among these factors, an increase in water temperature tends to enhance bacterial activity [66], which may have contributed to degradation. In Mangoku-ura, the seabed remained exposed for extended periods, potentially leading to the degradation of sedDNA due to UV radiation. In the first place, there was a possibility that the production of sedDNA from *U. major* during this period was limited. If degradation advances under such conditions, the likelihood of low concentrations increases.

Sanbanze experienced the highest water temperatures in July among all study sites, yet sedDNA at concentrations of approximately $10^3$ to $10^5$ copies/g sediment was extensively

detected. Furthermore, 30% of the observed burrow openings were Type 1, suggesting that some individuals may have been in a state of high physiological activity. This may be due to regional variations in the heat resistance of *U. major* populations. The hairy-handed crab *Hemigrapsus crenulatus*, a decapod that inhabits estuarine mudflats in New Zealand, exhibits variations in heat resistance among regions corresponding to temperature gradients with latitude [67]. Sanbanze, located in Tokyo Bay, is no stranger to water temperatures surpassing 30 °C during every summer in recent years. Hence, it is reasonable to posit that *U. major* inhabiting this area exhibits resilience to high water temperatures and maintains superior performance compared to its counterparts in Lake Akkeshi and Mangoku-ura. The production of sedDNA in the Sanbanze population did not decrease significantly. However, the number of detections decreased significantly in September. This indicated the involvement of a blue tide, which causes the appearance of hypoxic water masses along the coast. Between late August and late September 2023, there were widespread instances of blue tide events reported within the inner region of Tokyo Bay, where regions recorded dissolved oxygen (DO) levels at or below 0.5 ml/L [68]. According to an announcement from Chiba Prefecture, the blue tide around Sanbanze continued from September 22 to 26th, the day before the survey began. There are no direct studies on the response of *U. major* to blue tide, but it is known that thalassinideans, including this species, have lower oxygen consumption rates and are more tolerant to hypoxic conditions compared to typical decapods [69]. However, there are reports that this species exhibits a decrease in respiration rate and a significant reduction in feeding behavior under hypoxic conditions [70], and that tolerance decreases when dissolved oxygen (DO) falls below 2.5 mL/L [71]. *U. major* likely tolerated the hypoxia during the blue tide event, but it is unlikely that its physiological activity was high. This could have contributed to a decrease in sedDNA production.

## Conclusion

In three study sites with varying distribution of burrows and physical characteristics of tidal flats, the relationship between the sedDNA concentration and *U. major* abundance was investigated. As a result, in tidal flats with stable bottom environments, such as Lake Akkeshi and Mangoku-ura, it was found that areas with a higher number of burrow openings during the growth period of *U. major* exhibited significantly higher sedDNA concentrations. In other words, under these conditions, there was a correlation between the sedDNA concentration and abundance. However, it has become clear that concentrations can vary significantly owing to the presence of biological events, such as spawning periods, as well as changes in activity associated with environmental fluctuations. Conversely, in Sanbanze, which is marked by forceful wave dynamics and notable anthropogenic disturbances, no alignment between abundance and sedDNA concentration was evident during most months. These results underscored the importance of judiciously selecting the timing and sites for sediment sampling when using sedDNA as an abundance indicator. Further acquisition of fundamental data is imperative for establishing suitable sampling sites and seasons. Currently, locations characterized by minimal sedimentary environmental fluctuations, particularly those in bag-shaped bays, and periods without biological events are suitable for the sedDNA analysis of this species.

On the other hand, to estimate abundance with high accuracy from sedDNA concentration, it is essential to interpret the concentration data accurately. SedDNA concentrations can be regarded as a snapshot capturing a part of its production and degradation cycle. Understanding the production and degradation rates of sedDNA, while considering differences in sex, growth stages, and seasonal variations, will enable a more detailed interpretation of concentration data. Elucidating the dynamics of sedDNA is the next challenge and will likely be the key to unraveling the relationship between abundance and sedDNA concentration.

This study is the first to investigate the effectiveness and optimal conditions of sedDNA analysis for estimating the abundance of marine benthos, and is expected to serve as a foundation for future applied research.

## Supporting information

**S1 Fig. Verification of species-specificity of the primer-probe set using qPCR.** The organisms targeted for this assessment included *U. major*, its closely related species *U. yokoyai*, the dominant species at each survey site, *R. philippinarum*, *P. minutus* and *B. cumingii*. The number of samples is one for each species (one sample per group for *U. major*). The amplification was confirmed for only the four *U. major* samples from Groups 1 to 4.
(JPG)

**S1 File.** S1 Table. The total sedDNA concentration (ng/μL) of in each sample. BRA: burrow-rich areas, Lowercase letters (a–d) represent transects, and numbers indicate sampling locations (distance from 0 m point) for sediment. **S2 Table.** Results of the validation of the optimal annealing temperature in qPCR. Experimentation was conducted twice at temperatures of 60°C, 58°C, and 55°C, with the results indicating that 55°C yielded the highest detection count. **S3 Table.** The number of burrows per quadrat and the total number of burrow openings from 2021 to 2023. The blank areas indicate months where no survey was conducted. **S4 Table.** The numerical values indicating the reaction efficiency in qPCR. **S5 Table.** Copy numbers and sedDNA concentration data (Copies/g sediment). BRA: burrow-rich areas, Lowercase letters (a–d) represent transects, and numbers indicate sampling locations (distance from 0 m point) for sediment. **S6 Table.** Test results for sedDNA concentration between the burrow-rich area and other areas (A to D transects), as well as between each transect, using the Tukey–Kramer test (Q Value (α == 0.05)). **S7 Table.** The results of monthly sedDNA concentration comparisons within the burrow-rich area and across each transect, using the Tukey-Kramer test (Q value, α == 0.05).
(XLSX)

**S1 Appendix. Video of a sand volcano observed at Sanbanze.**
(MP4)

**S2 Appendix. Video of seawater spouting observed at Sanbanze.**
(MP4)

## Acknowledgments

We thank Mr. Satoru Takeyama, Director of the Akkeshi Oyster Seeding Center, and the Ishinomaki Bay Branch of the Miyagi Prefecture Fisheries Cooperative Association for their understanding of the field survey and cooperation in transporting the samples. We express our gratitude to Ms. Rumiko Kaneko and Ms. Makoto Suzuki for their assistance with the sedDNA experiments and morphological measurements. We would like to thank Editage (www.editage.com) for their English language editing. The comments provided by the two anonymous reviewers have greatly contributed to the significant improvement of our manuscript.

## Author contributions

**Conceptualization:** Kyosuke Kitabatake.

**Formal analysis:** Kyosuke Kitabatake, Natsuko Ito-Kondo.

**Funding acquisition:** Kyosuke Kitabatake, Kentaro Izumi.

**Investigation:** Kyosuke Kitabatake, Kentaro Izumi, Kenji Okoshi.

**Methodology:** Kyosuke Kitabatake, Natsuko Ito-Kondo.

**Supervision:** Kenji Okoshi.

**Writing – original draft:** Kyosuke Kitabatake.

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
