## [Decision Letter · Decision Letter 0]

12 Nov 2024

PONE-D-24-39393Sedimentary DNA is a promising indicator of the abundance of marine benthos: Insights from the burrowing decapod Upogebia majorPLOS ONE

Dear Dr. Kitabatake,

Thank you for submitting your manuscript to PLOS ONE. After careful consideration, we feel that it has merit but does not fully meet PLOS ONE’s publication criteria as it currently stands. Therefore, we invite you to submit a revised version of the manuscript that addresses the points raised during the review process.

**ACADEMIC EDITOR:**Reviewer's 1 reply to question 4 contradicts what is written in the first line of the review. The 'no' in question 4 was probably a mistake. I consider that the manuscript is presented in an intelligible fashion and written in standard English, despite needing the improvement mentioned by the reviewers.

We look forward to receiving your revised manuscript.

Kind regards,

Luísa Borges, PhD

Academic Editor

PLOS ONE

Journal requirements:    When submitting your revision, we need you to address these additional requirements. 1. Please ensure that your manuscript meets PLOS ONE's style requirements, including those for file naming. The PLOS ONE style templates can be found at https://journals.plos.org/plosone/s/file?id=wjVg/PLOSOne_formatting_sample_main_body.pdf and https://journals.plos.org/plosone/s/file?id=ba62/PLOSOne_formatting_sample_title_authors_affiliations.pdf 2. We note that the grant information you provided in the ‘Funding Information’ and ‘Financial Disclosure’ sections do not match.  When you resubmit, please ensure that you provide the correct grant numbers for the awards you received for your study in the ‘Funding Information’ section. 3. Thank you for stating the following financial disclosure:  [Part of this study was conducted with the support of JSPS KAKENHI Grant Number JP23K28209, the Sasakawa Scientific Research Grant from The Japan Science Society and The Research Institute of Marine Invertebrates.].  Please state what role the funders took in the study.  If the funders had no role, please state: ""The funders had no role in study design, data collection and analysis, decision to publish, or preparation of the manuscript."" If this statement is not correct you must amend it as needed. Please include this amended Role of Funder statement in your cover letter; we will change the online submission form on your behalf.

Reviewers' comments:

Reviewer's Responses to Questions

**Comments to the Author**

1. Is the manuscript technically sound, and do the data support the conclusions?

Reviewer #1: Partly

Reviewer #2: Yes

2. Has the statistical analysis been performed appropriately and rigorously? 

Reviewer #1: No

Reviewer #2: I Don't Know

3. Have the authors made all data underlying the findings in their manuscript fully available?

Reviewer #1: Yes

Reviewer #2: Yes

4. Is the manuscript presented in an intelligible fashion and written in standard English?

Reviewer #1: No

Reviewer #2: Yes

5. Review Comments to the Author

Reviewer #1: Manuscript Number: PONE-D-24-39393

Manuscript Title: Sedimentary DNA is a promising indicator of the abundance of marine benthos: Insights from the burrowing decapod Upogebia major

The MS seems likely to satisfy all of the criteria for publication in PLOS if it can be significantly revised. The article is written in excellent, standard English and presents the results of original research that has not been published elsewhere. The research also appears to meet all applicable standards for the ethics of experimentation and research integrity and the article adheres to appropriate reporting guidelines and community standards for data availability.

The experiments, statistics, and other analyses, however. are not performed to a high technical standard or described in sufficient detail. Kitabatake, for example, mention one period where detections were lower in non-shrimp areas than in abundant shrimp areas but they detected sedDNA nearly everywhere. They do not provide a simple graph of sedDNA detections and burrow or shrimp abundance to indicate an overall correspondence with abundance. (Wouldn’t the expected correlation be better with biomass?) Their conclusions therefore do not appear to be presented in an appropriate fashion to demonstrate they are supported by their data.

It is not clear to this reviewer how “The findings of this study” .. “demonstrate the effectiveness and optimal conditions of sedDNA analysis for estimating the abundance of marine benthos” and also “also suggested new potential in sedDNA for evaluating the physiological activity of benthos.”

Presumably, sedDNA is a function of production relative to degradation but they don’t they say that. These problems should be easy to fix. Kitabatake et al., for instance, could mention that better estimates DNA degradation (with distance [samples at greater distances from the shrimp beds] and time [repeated samples of the same sediments containing U. major DNA]) are still needed for determining abundance.

The burrow opening size/morphology information seems to be a different story and largely irrelevant. Except as estimates of abundance, those data seem to distract from the sedDNA detection story. Their explanation of how the burrow diameter or type is, or could be, related to sedDNA detection is confusing. Burrow diameter times burrow number could be crudely related to biomass but Kitabatake don’t use those data for that. Sediments from their tank experiment, described briefly, was not sampled for sedDNA. Samples were not held and tested over time to test for degradation of reductions of detection anywhere.

Minor comments:

Line 87-88: “in recent years, the quantitative assessment of this species has decreased.” What does this mean? Is there a reference?

Line 121: “U. major is a generalist” – this requires an explanation.

Line 226: sampled with - “sterilized plastic spoons soaked in 70% ethanol”. Ethanol preserves DNA. Explain how this avoided sample contamination. Was a different sterilized plastic spoon used for each sample?

Reviewer #2: This paper is a challenging attempt to use sedDNA as an indicator to examine the density of a large uopogebiid shrimp, which is an ecosystem engineer in tidal flats of the West Pacific. The results are not easy to interpret, as they are influenced by a number of factors, including the shape of the tidal flat, seasonal changes in the activity of the shrimp, and the dispersion and degradation rates of sedDNA. Even so, the authors have given honest and accurate consideration to each result, and there are no parts where the logic breaks down. Because the data was collected appropriately, the data published in this paper is expected to gradually increase in value as future research develops.

Please refer to the following on the method of displaying and analyzing the results.

L. 284. The statistical analysis methods should be in a separate section. It is also not good to have the details written as captions for supplementary tables. Since the presence or absence of seasonal and spatial changes is mentioned in the main text, please describe the analysis methods in the main text.

Perhaps Figures 5 and 6 are the most important results, but you need to explain in the Methods section what you are comparing there, and you also need to make the figures of the results easy to understand.

In particular, when it comes to seasonal changes, it is only appropriate to compare the burrow-rich area, and this should be shown in a bar graph like Figure 6. When I only look at Table S7, it looks as if there are no seasonal changes.

By the way, isn't the order of the letters of transect ABCD in Figure 2b wrong?

I think it doesn't match up with Figure 5.

L. 409. There is no description in the Methods section about the behavioral observation. If you are going to describe the results, you need to describe the methods. However, even if these behaviors or activities were not observed at low tide, it is not possible to deny the possibility that they are carried out at high tide, so I think that this is an extremely limited observation that supports the authors' discussion.

The description from L. 513 to L. 526 cannot be used in the discussion unless it is described in the methods and results. I think it would be good to summarize the methods and observation in the supporting information.

The following are important reviews of the ecology of the upogebiid shrimps, so please add them to your citations.

Atkinson, R. J. A. and A. C. Taylor (2005). "Aspects of the physiology, biology, and ecology of thalassinidean shrimps in relation to their burrow environment." Oceanography and Marine Biology: An Annual Review 43: 173-210.

Kinoshita, K. (2022) “Life history characteristics and burrow structure of the mud shrimp (Decapoda: Upogebiidae)” Plankton and Benthos Research 17: 327-337/

6. PLOS authors have the option to publish the peer review history of their article (what does this mean? ). If published, this will include your full peer review and any attached files.

**Do you want your identity to be public for this peer review?** For information about this choice, including consent withdrawal, please see our Privacy Policy .

Reviewer #1: No

Reviewer #2: No

---

## [Author Response · Author response to Decision Letter 1]

28 Dec 2024

Dear Dr. Luísa Borges:

Thank you for giving us to submit a revised draft of our manuscript entitled, “Sedimentary DNA is a promising indicator of the abundance of marine benthos: Insights from the burrowing decapod Upogebia major” to PLOS ONE.

We greatly appreciate the time and effort you and the reviewers have dedicated to providing insightful feedback to strengthen our paper. It is with great pleasure that we resubmit our article for your further consideration. We have incorporated changes that reflect the detailed suggestions you graciously provided. We hope that our revisions and responses satisfactorily address all the comments and concerns raised by you and the reviewers.

Sincerely,

Kyosuke Kitabatake*, Kentaro Izumi, Natsuko Ito-Kondo, Kenji Okoshi

*Faculty and Graduate School of Education, Chiba University, 1-33 Yayoi, Inage, Chiba, Chiba 263-8522, Japan

---

## [Editor Report · Decision Letter 1]

14 Jan 2025

Sedimentary DNA is a promising indicator of the abundance of marine benthos: Insights from the burrowing decapod Upogebia major

PONE-D-24-39393R1

Dear Dr. Kitabatake,

We’re pleased to inform you that your manuscript has been judged scientifically suitable for publication and will be formally accepted for publication once it meets all outstanding technical requirements.

Kind regards,

Luisa Borges, PhD

Academic Editor

PLOS ONE

Additional Editor Comments:

Line 202- please delete 'slim'; giving the dimensions of the tanks is enough.

Line 203- It should be 'were used' instead of 'was used' for subject verb agreement;  please delete 'as the experimental tank', in line 202 you mentioned three tanks.

Line 2027- is 'chemical pipette' a special type of pipette? If it isn't, please delete 'chemical'; if it is, please provide details.

Line 678- please substitute 'enables' with 'will enable' to show clearly this is will de bone in future work.

---

## [Editor Report · Acceptance letter]

PONE-D-24-39393R1

PLOS ONE

Dear Dr. Kitabatake,

I'm pleased to inform you that your manuscript has been deemed suitable for publication in PLOS ONE. Congratulations! Your manuscript is now being handed over to our production team.

Kind regards,

on behalf of

Dr. Luisa Borges

Academic Editor

PLOS ONE